# Changes in physiotherapists' perceptions of evidence-based practice after a year in the workforce: A mixed-methods study

**Maureen McEvoy**[1]*, **Julie Luker**[1], **Caroline Fryer**[1], **Lucy K Lewis**[2]

**1** Allied Health and Human Performance Unit, University of South Australia, Adelaide, Australia, **2** Caring Futures Institute, College of Nursing and Health Sciences, Flinders University, Adelaide, Australia

These authors contributed equally to this work.

* maureen.mcevoy@unisa.edu.au

**Data Availability Statement:** All relevant data are within the manuscript and its Supporting Information files.

## Abstract

### Background

Few studies have explored evidence-based practice (EBP) knowledge, attitudes and behaviours of health professional graduates transitioning into the workforce. This study evaluated changes in these EBP domains in physiotherapists after one year of working.

### Method

A mixed methods design was used. Participants completed two psychometrically-tested EBP questionnaires at two timepoints. The Evidence-Based Practice Profile questionnaire collected self-report EBP data (Terminology, Relevance, Confidence, Practice, Sympathy) and the Knowledge of Research Evidence Competencies collected objective data (Actual Knowledge). Changes were calculated using descriptive statistics (paired t-tests, 95% CI, effect sizes). Qualitative interview data collected at one timepoint were analysed using a descriptive approach and thematic analysis, to examine the lived experience of participants in the context of their first employment. The aim of the mixed methods approach was a broader and deeper understanding of participants' first year of employment and using EBP.

### Results

Data were analysed from 50 participants who completed both questionnaires at the two timepoints. After one year in the workforce, there was a significant decrease in participants' perceptions of Relevance (p<0.001) and Confidence with EBP (p<0.001) and non-significant decreases in the other domains. Effect sizes showed medium decreases for Relevance (0.69) and Confidence (0.57), small decreases in Terminology (0.28) and Practice (0.23), and very small decreases in Sympathy (0.08) and Actual Knowledge (0.11). Seven themes described participants experience of using EBP in their first working year.

**Funding:** MM, LL, JL received a University of South Australia Learning and Teaching Grant 2013 ($9985.23) (no number). The funders had no role in study design, data collection and analysis, decision to publish, or preparation of the manuscript

**Competing interests:** The authors have declared that no competing interests exist.

## Conclusions

After a year in the workplace, confidence and perceptions of relevance of EBP were significantly reduced. A subtle interplay of features related to workplace culture, competing demands to develop clinical skills, internal and external motivators to use EBP and patient expectations, together with availability of resources and time, may impact early graduates' perceptions of EBP. Workplace role models who immersed themselves in evidence discussion and experience were inspiring to early graduates.

## Introduction

Since the introduction of evidence-based practice (EBP) in medicine in the early 1990's [1], many professions allied to health and social care have embraced an evidence-based approach to learning and clinical practice [2]. The five-step model of EBP [3] is commonly accepted as core curricula in entry-level health professional training, as evidenced by the inclusion of EBP in accreditation documents for health professional programs to enable registration of graduates in their chosen professions [4].

While there are many quantitative studies investigating the effectiveness of EBP teaching in medicine and allied health, more recently, mixed model studies have emerged, providing for richer exploration of EBP. These studies have been in undergraduate-entry level students [5–7] and in clinicians with varying experience [8–10]. The transition period from completion of health professional training to workplace practice is an area of sparse research. A recent mixed methods study was limited by the small number of graduates (n = 13) included in a cross-sectional comparison across year levels in a graduate-entry occupational therapy program [11]. It is well accepted that training entry-level students in EBP results in improved self-reported knowledge and attitudes, a sound understanding of the EBP process and positive attitudes for its application in clinical practice, on graduation [12]. McEvoy et al. [13] also provided quantitative data on changes in EBP knowledge, attitudes and behaviour after transitioning into the workplace for one and two years. However, there was no further exploration of what may influence these changes.

There are likely to be many factors impacting new graduates' decision making and use of EBP in the workplace including time, access, workplace culture and resources [14]. Workplace dissemination and implementation of EBP may be a complex process relating to the clinician, workplace environment and culture [14], and quality of previous training [15]. However, a systematic review by Beidas et al. [16] reported that there was insufficient information about how clinician, patient and organisational variables influence EBP competence, skill and adherence in the workforce.

Examining EBP competence and behaviours on transition from entry-level health professional programs to the workplace can inform EBP education and practice across professional training years. Recommendations are needed for how best to prepare and support EBP during clinicians' early years in the workplace. Therefore, the objective of this study was to determine what is influencing the use of EBP by physiotherapists during their first year in the workforce. The specific aims of the study were to quantitatively examine overall changes in domain scores associated with EBP knowledge, attitudes and behaviours, and to qualitatively explore the experience of using EBP in a cohort of physiotherapists after a year in the workforce.

## Methods

This paper describes the second stage of a larger study where physiotherapy students' EBP knowledge, attitudes and practices were tracked during their entry-level physiotherapy

training, from baseline to graduation [12]. In the second stage, participants were followed up after their first year in the workforce. Data were collected from 2013 to 2014. Ethical approval was gained from the University of South Australia Human Research Ethics Committee (protocol numbers 0000021077 and 0000030567). All participants provided written informed consent.

## Design

Quantitative and qualitative procedures of the study were conducted independently of each other in a convergent mixed methods design [17]. The timepoint 1 quantitative data were collected on participants' completion of the final EBP course in their final year of the Physiotherapy Program. The timepoint 2 quantitative data were collected after participants had been in the workforce for up to 1 year (September-November 2014). The qualitative interviews were conducted in the immediate period after the timepoint 2 quantitative data were collected (November 2014). The interview questions were independent of the quantitative results and not all participants who completed the survey, consented to the interviews. The quantitative and qualitative data were analysed separately then merged in the interpretation and reporting of results.

A rigorous descriptive qualitative approach was used with semi-structured interviews and thematic analysis [18]. Within an interpretive paradigm, this approach uses low-inference interpretation to provide a rich description of experiences and perspectives in everyday language [19]. A qualitative descriptive method was chosen to provide a pragmatic way of examining the lived experiences of our participants in the context of first employment, with a relativist ontology and a subjectivist epistemology [19, 20]. The two qualitative researchers who led this component of the study (CF, JL) are experienced physiotherapists and qualitative researchers in the field of health science at post-doctoral level.

## Participants

The target population was all physiotherapy students completing their final year of training at the University of South Australia in 2013 and entering the physiotherapy workforce in 2014 (n = 125). For inclusion, students needed to have completed all final year courses in late 2013 or early 2014, and to be entering the physiotherapy workforce (i.e. not continuing to further study, not delaying practicing as a physiotherapist). International students leaving Australia were excluded. A minimum number of 34 participants was needed for the quantitative data set in order to obtain 80% power for a two-tailed distribution, with alpha set at 0.05 and a medium effect size (d = 0.5) derived from the study of McEvoy et al [13].

## Procedure

**Quantitative data collection.** Quantitative data were collected using two questionnaires, the Evidence-Based Practice Profile (EBP$^2$) questionnaire and the Knowledge of Research Evidence Competencies (K-REC) questionnaire. The EBP$^2$ questionnaire has been psychometrically-tested [21] and shown to be valid and reliable (test-retest reliability: ICC 0.77 to 0.94, internal consistency: Cronbach's alpha 0.96, convergent validity: Pearson r = 0.54–0.80, can distinguish between groups for different levels of EBP exposure $p<0.05$). The EBP$^2$ questionnaire includes 58 items (5-point Likert scale) that relate to five self-reported EBP domains: *Relevance* (14 items) is the value, emphasis or importance placed on EBP; *Terminology* (17 items) relates to an understanding of common research terms; *Confidence* (11 items) is perception of ability in EBP skills; *Practice* (9 items) is the application of EBP in clinical decision-making; and *Sympathy* (7 items) is the perceptions of compatibility of EBP with day-to-day professional

work [21]. The K-REC questionnaire has 9-items that measure participants' knowledge of EBP using multiple choice, true and false and an open-ended question, pertaining to a clinical scenario, and has been demonstrated to be a valid and reliable tool (test-retest reliability: Cohen's Kappa and ICC range 0.62 to perfect agreement, can distinguish between groups with and without formal EBP training $p < 0.0001$) [22].

Participants completed the questionnaires at two timepoints: 1) at completion of entry-level training, Timepoint 1, and 2) one year after entering the workforce, Timepoint 2. At Timepoint 1, participants answered the questionnaires using a pen-and-paper format. For Timepoint 2, participants answered using an electronic version of the questionnaires through Survey Monkey, due to the participants being geographically dispersed. All participants were allocated a unique identifier to allow matching of data from both timepoints. Follow-up reminders were sent to those who did not open their individual link to the electronic questionnaire.

**Qualitative data collection.** Qualitative data were collected after one year in the workforce using telephone interviews. A semi-structured interview guide was developed by the research team independent of the quantitative data findings. Content validity of the interview guide was supported by seeking feedback during its development from a reference group of six physiotherapists with broad EBP teaching experience, including a researcher in this field (S1 Appendix). Topics covered by the interview questions included: how EBP is used in the workplace, the relevance and value of EBP in the workplace, the impact of EBP role models, how EBP workplace behaviours are encouraged, and usefulness of EBP training to prepare for graduate practice. An interviewer, independent of the research team and experienced in qualitative interviewing, conducted telephone interviews with the participants. Each interview was recorded then transcribed verbatim by a professional service. Transcript data were de-identified and entered in NVivo 10 software for analysis.

## Data management and analysis

**Quantitative analysis.** De-identified data were entered in Predictive Analytic Software (PASW) Statistics 17.0 (Chicago, IL). Participants' scores for the domains were only included if matched data for the domain were available for the two timepoints. If all items in a domain on either occasion were not competed, this domain score was not included in the analysis. The Likert scores for the EBP$^2$ questionnaire were treated as interval data. The maximum domain scores varied (Relevance 70, Terminology 85, Confidence 55, Practice 45, Sympathy 35), due to the different number of items per domain. The K-REC instrument gives a maximum score of 12. Participant responses were scored using set scoring guidelines. Descriptive statistics were calculated for each of the five EBP$^2$ domain scores, the K-REC domain total score and demographic information. Paired t-tests, 95% CI and effect sizes (ES) were used to examine changes between the two timepoints. The ES were classified as very small (0.01), small (0.20), medium (0.50) and large (0.80) [23]. Alpha was set at 0.05. As maximum domain scores varied due to the different numbers of items, the scores were calculated as a percentage of possible maximum (100%) at each timepoint, for graphical presentation.

**Qualitative analysis.** Two researchers (CF and JL) separately analysed the qualitative data using a staged process of thematic analysis. First researchers read the transcripts repeatedly to familiarize themselves with the data. Second, codes were inductively allocated to small sections. Third, comparison across transcripts and codes were done, inductively grouping the codes in a meaningful manner to form categories and themes within each category [24]. This was an iterative process requiring the researchers to meet and discuss regularly to eventually reach a consensus.

The quantitative and qualitative findings of the study were integrated narratively to address the study objective [17]. The quantitative findings are presented first then the subsequent reporting of qualitative findings uses a weaving approach to integrate the qualitative themes with the quantitative domain results.

## Results

### Participants

Fig 1 illustrates the flow of participants through the study. Of the 125 eligible participants, 87 had complete questionnaire data at timepoint 1 [12]. Of these, 84 participants consented for follow-up questionnaires at timepoint 2 after one year in the workforce, but 19 were ineligible for follow-up (failed to successfully complete final courses in the physiotherapy program, planning to work overseas or not to work in physiotherapy in the first year after graduation), resulting in 65 eligible participants for timepoint 2. Of these, 50 (70%) participants completed the questionnaires at timepoint 2. Of these 50 participants, 14 participated in interviews after timepoint 2 completion of questionnaires.

At timepoint 1, of the 50 participants there were 35 females and 15 males. The mean age was 22 +/- 4 years (range 20–47). At timepoint 2, 14 participants were interviewed. There were 10 females and 4 males, with an average of 22 +/- 1 years (range 20 to 23). The interviewees worked in a range of settings in hospitals (n = 4), private practice (n = 7) and community health (n = 3), in both metropolitan (n = 9) and regional (n = 5) areas. Most interviewees treated both inpatients and outpatients ± aged care (n = 7), while others treated outpatients only ± sports (n = 5) and inpatients only (n = 2). While 50 participants completed both

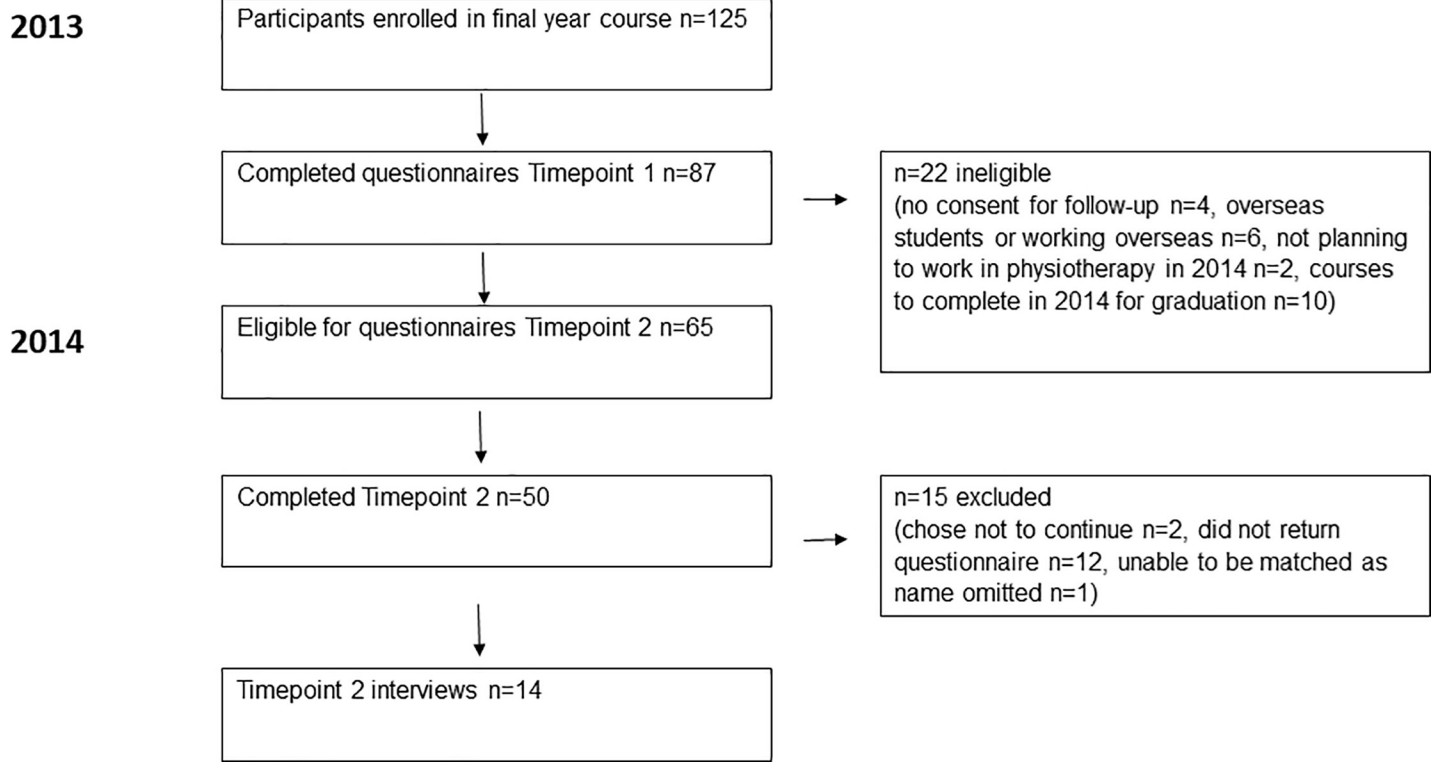

**Fig 1. Flow of participants through the study.**

**Table 1. EBP domain scores, change, *p* values and effect sizes for participants at completion of entry level training and after one year in the workforce (n = 50)\*.**

| EBP domain | Timepoint 1 mean (SD) | Timepoint 2 mean (SD) | Change (95% CI) | Raw p values | Effect size (ES) |
|---|---|---|---|---|---|
| Relevance n = 49 (max. score 70) | 64.9 (4.4) | 60.9 (5.4) | 4.0 (2.4 to 5.7) | ***p*<0.001** | ES 0.69↓ |
| Terminology n = 49 (max. score 85) | 66.6 (9.1) | 64.2 (11.0) | 2.4 (-0.0 to 4.8) | *p* = 0.055 | ES 0.28↓ |
| Confidence n = 49 (max. score 55) | 44.6 (6.0) | 40.9 (7.4) | 3.7 (1.8 to 5.5) | ***p*<0.001** | ES 0.57↓ |
| Practice n = 50 (max. score 45) | 26.9 (5.6) | 25.5 (5.6) | 1.4 (-0.35 to 3.3) | *p* = 0.11 | ES 0.23↓ |
| Sympathy n = 48 (max. score 35) | 24.0 (3.5) | 23.7 (3.0) | 0.3 (0.80 to 1.4) | *p* = 0.59 | ES 0.08↓ |
| Actual Knowledge n = 49 (max. score 12) | 8.9 (1.8) | 8.7 (1.2) | 0.2 (-0.35 to 0.81) | *p* = 0.43 | ES = 0.11↓ |

*Complete data was not provided for all domains by the 50 participants; Timepoint 1: completion of entry-level training; Timepoint 2: one year in the workforce; Bolded *p* values are statistically significant (p<0.05)

questionnaires at the two timepoints, there was not complete matched data from all participants for each of the six EBP domains (Practice n = 50; Relevance, Terminology, Confidence and Actual Knowledge n = 49; Sympathy n = 48). The domain structure of the questionnaire allowed for this analysis provided all items in a domain were completed.

## Quantitative

Table 1 presents the domain scores (mean, change, CI, p values and effect sizes) for the 50 participants with matched data at the two timepoints. After one year in the workforce, there was a significant decrease in participants' perceptions of the Relevance of EBP (*p*<0.001) and self-reported Confidence with EBP (*p*<0.001). There were non-significant decreases in the other domains of Terminology, Practice, Sympathy and Actual Knowledge. In terms of effect sizes, there were medium decreases in the Relevance and Confidence domains, small decreases in Terminology and Practice, and very small decreases in Sympathy and Actual Knowledge.

The maximum domain scores in the EBP[2] questionnaire vary due to the different numbers of items in each domain. Fig 2 presents the domain scores as a percentage of the possible maximum (100%) score in each domain across the two timepoints.

## Qualitative

Three categories were developed from the descriptive analysis of interview data: 'Using EBP in the first year', 'The EBP undergraduate course', and 'First year experience'. The category 'Using EBP in the first year' is reported in the current paper. The second category 'The EBP undergraduate course' was used to inform changes to the program curriculum. The final category 'First Year experience' described findings relevant to a broader context than the topic of this paper and will be published separately.

The experience of participants 'Using EBP in the first year' is described by the following seven key themes:

1. Looking up research evidence when unsure

2. Clients appreciate hearing about evidence for their treatment

3. Workplace culture supports EBP behaviours

4. Difficulties accessing time and databases for EBP

5. Confidence in EBP skills comes with practice

6. Learning a lot from colleagues' clinical expertise

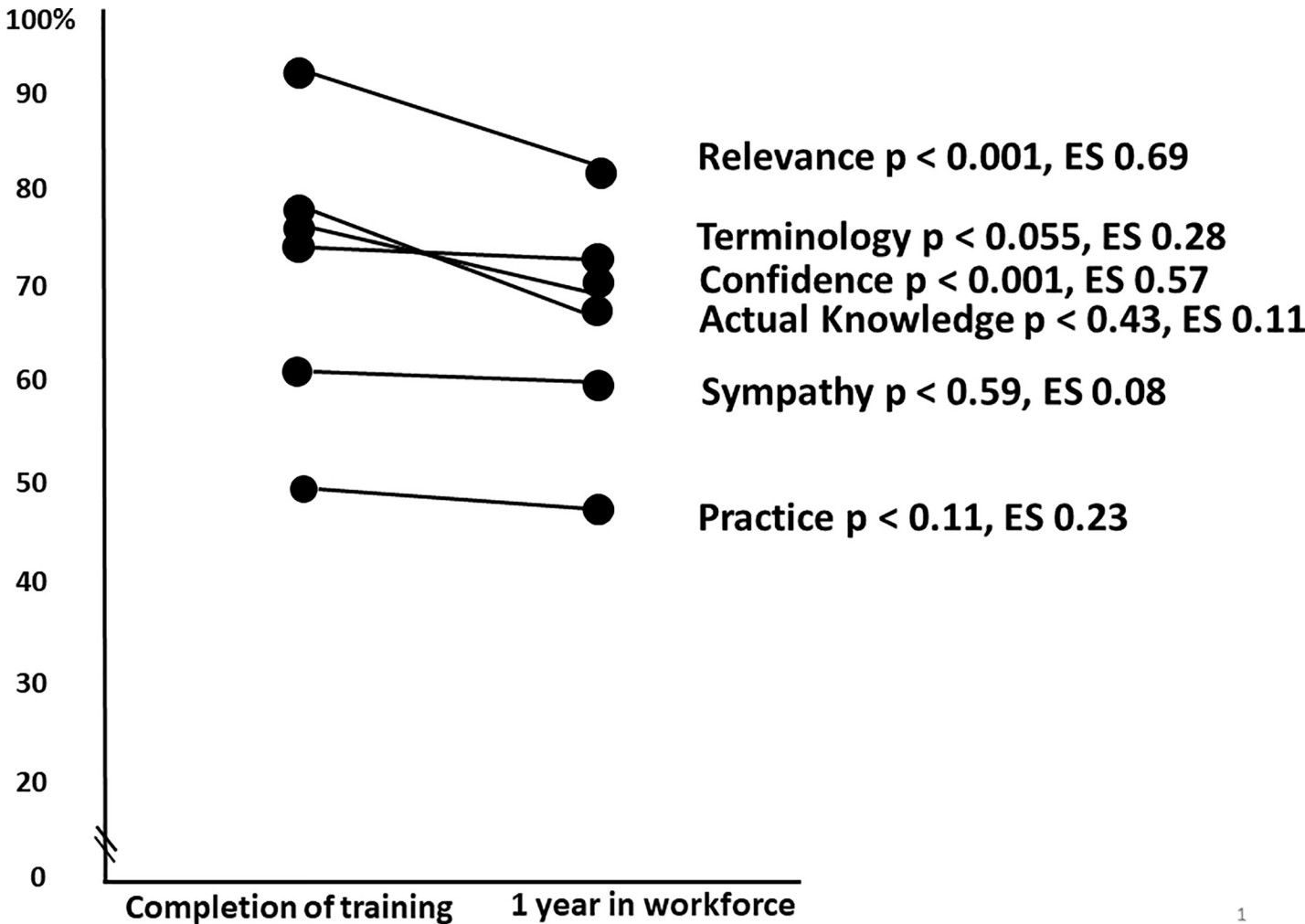

**Fig 2. Domain scores from the EBP2 questionnaire shown as a percentage of the possible maximum score for each domain, across the two timepoints.**

7. Self-motivated to be EBP user

Each theme is presented narratively with illustrative data quotes [18] and integrated with the quantitative results.

**Theme 1: Looking up research evidence when unsure.** Participants' primary application of EBP skills and knowledge in their first working year was to search for research evidence to inform clinical reasoning when they were unsure about what to do. This was usually in the context of treatment decisions for individual clients as Participant 7 explained,

'You want to make sure, if someone comes in with a condition you're not too sure about or has had something done, then you'll look it up in response to that',

but was also used by a few participants to develop intervention programs for specific clinical populations,

'We came up with a program for hamstring training, so we seemed to be getting a fair few come through, so I had to pretty much go back and do almost like a Uni-type assignment' (participant 44).

Only three participants reported using their own clinical expertise and incorporating the patient's goals and views in addition to the research evidence in their reasoning,

> 'I obviously don't rely heavily on what it says in the literature all the time. I take to mind what it says and my clinical experience and the patient goals—like the three bits of EBP' (Participant 14).

This finding that participants regularly searched and appraised evidence in their first year supports the maintenance of participants' performance in the quantitative 'Practice' domain.

**Theme 2: Clients don't ask but appreciate hearing about evidence for their treatment.** Participants generally perceived that clients do not have much idea about EBP and do not ask for evidence for their treatment. Yet most participants believed that clients appreciated being told about research evidence as it encouraged them and gave them confidence in the physiotherapy treatment. Participant 56 explained,

> 'I've had a few patients that they may have been a little bit apprehensive about some of the treatments that we use, but then we sort of explain to them about the evidence behind it and it really does help them to feel confident in what we're doing and that we're not just doing some sort of weird treatment.'

Interest in research evidence was noted to differ between client groups, with greater interest perceived in clients with a health background such as personal trainers. This finding that patients do not ask for evidence may have influenced the reduction in the quantitative domain of 'Relevance'.

**Theme 3: Workplace culture supports EBP behaviours.** The role modelling of an EBP approach by work colleagues, and direct encouragement by supervisors to be evidence-based were key extrinsic motivators for participants to use their own EBP skills. Regular in-service education and professional development were perceived as helpful to develop EBP knowledge. Frequent informal discussions of recent evidence between colleagues and widespread sharing of journal articles in the workplace strongly encouraged participants to use EBP. The following two quotes illustrate this experience,

> 'They're all role models, because every time they do something, or I see them reading an article, it kind of reminds me that, oh yeah I should do that, or I haven't done that in a while.' (Participant 12)

> 'At my clinic work they view EBP obviously quite high in a sense that they always want to find out whether you have an opinion and where's the evidence to that opinion and where's the evidence to back it up.' (Participant 22)

Nine participants identified key role models, often in high-ranking positions, who worked in or visited their workplace and inspired them with dedication to conducting and using research evidence. In contrast, three participants experienced older colleagues who demonstrated disinterest in EBP by failing to update their practice. Mostly this was not perceived to be a barrier to participants' own EBP behaviour,

> 'I guess they're stuck in the stuff that they were taught when they were at Uni still. They use a lot of EPAs and stuff like that, which I'm not 100% keen on.' (Participant 14).

but one participant experienced it as actively discouraging their use of EBP,

'. . .we just have a couple of senior physios and my perception is that they're more inclined to sort of go with, like their clinical experience and so sometimes that can get a little bit, "Well we know this, like so just do this," rather than spend that time looking it up. (Participant 62).

The strength of an EBP culture was perceived to differ between clinical areas by participants who worked across areas in their first year. Outpatient rehabilitation, intensive care and private musculoskeletal practice were viewed as areas where there was a lot of research evidence available and used. Aged care, hospital inpatients and paediatrics were identified as less relevant for EBP as treatment techniques were perceived as more restricted and repetitive. The finding of regular in-service education and sharing of evidence between colleagues in workplaces supports participants' strong performance in the quantitative 'Terminology' domain for common research terms. Perceived differences in the use of EBP between clinical areas may have contributed to the reduction in the quantitative domain of 'Relevance'.

**Theme 4: Difficulties accessing time and databases for EBP.** Key barriers to participants using their EBP skills and knowledge were difficulty finding time to access research evidence and lacking access to online databases or journals at their workplace. A lack of time was associated with full patient loads, understaffing and the absence of dedicated time for EBP practice,

'I think sometimes when we're busy we're just trying to get everything done and just get it done quickly and we don't have time to sit down and go through that stuff or think about that stuff. I think that's one of the main reasons it doesn't get used sometimes.' (Participant 49).

Access to online database access was not available at many workplaces or the participants had not been orientated to the process of gaining access,

'It was kind of hard, like I felt like even if I wanted to, I didn't really know where to go for it. But I guess they weren't actively discouraging it, it was just no-one knew about the access.' (Participant 12).

The impact of these barriers on participants use of skills is reflected in the reduced quantitative domain of 'Confidence' in ability to use their skills.

**Theme 5: Confidence in EBP skills comes with practice.** Many participants reported a reduction in their confidence to use EBP skills and knowledge during the year. Confidence in EBP skills was strongly associated by participants with the amount they were able to practice the skills they had learned during their university training. A lack of practice in using statistical terms and critically appraising evidence was specifically highlighted

'I think it's probably gone down a bit because I'm not constantly refreshing and critiquing articles. I probably look up an article or two every week but it's more I'm not being very critical of it.' (Participant 40)

The few participants who reported they gained confidence in their use of EBP variously identified this to be associated with attending presentations, observing others using evidence, and putting their EBP training into practice. This finding suggests a lack of regular practice of all EBP knowledge and skills contributed to the reduced quantitative domain of 'Confidence' in ability to use their skills.

**Theme 6: Learning a lot from colleagues' clinical expertise.** Nine of the 14 participants reported seeking advice from senior colleagues about patient treatment. They valued the clinical expertise of their more experienced colleagues about 'what works well' (Participant 37). The advice received from senior colleagues was considered by participants in addition to, and sometimes in preference to, research evidence. Participant 56 explained this experience,

'I've got some very experienced physios working where I am, so I am sort of using their experience as well and hearing what they've got to say. Sort of asking questions from them. They've been really good as well and being able to sort of look at the evidence and go yeah that's great but for this patient it may not necessarily work for these reasons.'

This finding that participants frequently sought advice from colleagues based on clinical experience may have reduced the comparative perceived importance of using research evidence, therefore reducing the quantitative domain of 'Relevance'.

**Theme 7: Self-motivated to be EBP user.** The majority of participants expressed an intrinsic motivation to use EBP to become a 'good' and effective physiotherapist. This self-motivation was present in participants who did and did not work within a strong EBP culture. Participant 44 insisted their intrinsic motivation was independent of external influences to use EBP,

'I think I've got a pretty high drive to use it anyway. Particularly like that, if you want to be good at what you are doing, you try to take that on board, you can't really ignore EBP and all that stuff.' (Participant 44).

This finding that self-motivation to use EBP was strong within the group may explain the maintained level of the 'Practice' domain in the quantitative results.

## Discussion

The key findings of this study were that physiotherapy graduates after one year in the workforce reported small reductions in knowledge, practice and sympathy for EBP, and significant declines in confidence and perceptions of the relevance of EBP compared to these measures at graduation. These quantitative domain declines appeared from qualitative results to be influenced by a range of inter-related influences, including workplace culture and the opportunity to practice EBP skills.

Relevance of EBP in clinical practice showed an overall decline after one year in the workforce but the qualitative findings suggest that this varied according to area of clinical practice. Participants reported that EBP had greatest relevance in the management of musculoskeletal conditions where presentations were varied and progressed through stages of rehabilitation. There was a perception of greater autonomy in management and integration of best research evidence in clinical reasoning. By comparison, participants felt the value and use of EBP in other clinical areas was limited by little or slow changes in practice. All clinical areas in physiotherapy are actively researched and have research evidence available to clinicians. It is possible that the large quantity of musculoskeletal research in physiotherapy conference programs and many journals influenced the perception of relevance, rather than the quantity or quality of the research actually available. It may also be that wider health care systems impact on physiotherapists' practice and ability to use EBP e.g. where funding models direct care. Furthermore, the perceived relevance of EBP appeared to be influenced by competing demands for early career physiotherapists to learn from experienced colleagues and improve clinical skills. Relevance may also have been impacted by the perceived lack of expectation regarding EBP from

patients. However, when evidence is explained, many patients appreciate the decision-making underpinning treatment choices and communicating evidence needs to be recognised by practitioners as a professional obligation relevant to the communication requirements of informed consent [25].

The overall decline in the Confidence domain identified in this current study may have been influenced by participants reported difficulties in accessing the time and database evidence for practicing EBP. Participants reported retaining skills for searching databases, but were generally less critical of research with poorer ability to interpret the statistical measures and the quality of the research. A lack of consistent practice and a loss of knowledge in skills for accessing, appraising and integrating best research evidence were cited as contributors to having less confidence. Interestingly, following an initial decline after a year of working, McEvoy et al [13] reported improved confidence scores after two years of working. The authors hypothesised that early graduates may need an initial period to establish workplace relationships and clinical routines before their focus and confidence returns to EBP.

A decline in confidence in *clinical practice* in the first year of practice in physiotherapy has also been previously reported [26], with a greater 'outward' focus and confidence in the second year of practice. When their EBP skills are viewed in the context of the five stages of learning, graduates in the current study may have moved through novice and advanced beginner levels during EBP training across the physiotherapy program, and enter the workforce at a competent level [27]. Coaching and a balance of supervision and autonomy, with encouragement to self-reflect and justify decisions may have supported achievement of this level of EBP skills [28]. To reach the fourth level of proficiency in EBP skill, there needs to be support in decision-making and self-reflection in relation to more complex experiences to build confidence [28]. This level, which is characterised by taking responsibility for one's decisions, identifying opportunities to teach, and being responsible to others, may be stalled in the first year of working with the potential to re-emerge after two years as a working physiotherapist.

The results of this study indicate that a workplace culture that values research evidence strongly supports EBP use and helps in maintaining confidence and supporting EBP practice in the workplace. Some of the practices that cultivate this culture can be implemented in all workplaces, including ongoing discussions of research evidence, sharing of journal articles, regular in-service education, and the support and encouragement by supervisors to make evidence-based decisions. Role models within and outside the workplace who championed and practiced EBP were inspiring to new graduates and can be considered key motivators to using EBP in the first year of work.

There are no known previous studies that have evaluated the changes in EBP knowledge, attitudes and behaviours in this transitional period from graduation to a year in the workplace. Several studies have suggested that EBP training should be integrated across the professional continuum rather than constrained to entry-level programs prior to graduation and registration [29, 30]. Relating to EBP in psychology, Leffler et al [29] proposed individualised training and active strategies that are supported and have measurable outcomes of learning and effects. Examples of these approaches during training, during internship and for clinicians were provided. While not always available, internship is closest to the first year of practice. Leffler et al [29] proposed a dedicated period to train psychologists to integrate clinical research and practice, by the collaborative development of projects under the supervision of faculty members who engage in clinical research. Tilson and colleagues [8–10] developed a program *for clinicians* to support integration of research evidence into clinical decision-making. In the Physical therapist-driven Education for Actionable Knowledge translation program (PEAK), opportunities are provided for expert and peer support to explore new skills, using small group learning with an experienced EBP expert, librarian and with peers [9]. The PEAK program

improved short term self-reported EBP confidence and behaviours, but this was not sustained at six months [10]. Tilson et al [9, 10] did not report on whether participants had exposure to EBP during physiotherapy training. McEvoy et al [12] found significant improvement in self-reported EBP knowledge, attitudes and behaviours prior to graduation, after exposure to EBP courses. These improvements in the same cohort after graduation were not sustained, as reported in the current study. Lessons may be learned from Tilson et al [9, 10], to build further on a strong EBP foundation gained prior to graduation.

Recommendations from this research relate to clinical practice and education. A practice recommendation for the future may be to incorporate greater organisational support, with more monitoring, feedback and problem-solving offered in the transition period using EBP experts, librarians and peers. In addition, building stronger links between workplaces and physiotherapy training institutions may allow the mutual strengths of both to be better utilised. For example, students often have research and analysis skills to share, but need consolidation of clinical skills that can be provided by experienced clinicians. Together there can be integration of clinical and research evidence along with evidence from the patient, to provide authentic EBP.

Education recommendations may be to target statistics training focussing on *application* of statistics, to build confidence in this area. This should not be at the expense of continued development of all EBP skills, as entry-level education resulted in significant improvements in all domains [12]. A further recommendation may be to better prepare students for the challenges of the transition period where workplaces may provide limited support, and to encourage new graduates to seek role-models and opportunities to maintain EBP skills.

There are limitations to acknowledge in our study. Firstly, the study was undertaken in one institution and in a single profession potentially limiting generalisability. Secondly, it is possible that there were confounders such as place of employment, setting, rural, metropolitan and remote locations. As the primary aim of the quantitative arm of this study was to assess the change in domain scores after a year in the workforce, these variables were not assessed or included in the analyses. This is a possible area for future research.

Future mixed-method studies may be undertaken to collect data beyond the first year in the workplace where further changes in EBP knowledge, attitudes and behaviour may be explored. Studies exploring how graduate EBP knowledge, attitudes and behaviours may contribute to a successful workplace may also be of interest. Collecting data on access to resources, availability of professional development, role models and mentors to enhance EBP and further investigation of aspects of an EBP-supportive workplace culture may be valuable.

In conclusion, one year into working, physiotherapists showed a decline in all EBP profile domains, most significantly for confidence and the perceived relevance of EBP. In the first year of work, a subtle interplay of features related to workplace culture, competing demands to develop clinical skills, internal and external motivators to use EBP and expectations of patients, together with availability of resources and time, may impact early graduates' perceptions of EBP. Role models in and outside of the work environment who immersed themselves in evidence-based discussion and experience were identified by early graduates as inspiring.

## Supporting information

**S1 Appendix. Semi-structured interview guide.**
(DOCX)

**S2 Appendix. Quantitative dataset.**
(PDF)

**S3 Appendix. Coded interview dataset.**
(XLSX)

## Acknowledgments

The authors would like to acknowledge and thank statistician Alvin Atlas (International Centre of Allied Health Evidence, iCAHE, University of South Australia) for his assistance with data management and statistical analysis. The authors would like to thank the participants who were transitioning into the workforce for engaging in this research.

## Author Contributions

**Conceptualization:** Maureen McEvoy, Julie Luker, Lucy K Lewis.

**Data curation:** Maureen McEvoy, Caroline Fryer.

**Formal analysis:** Maureen McEvoy, Julie Luker, Caroline Fryer, Lucy K Lewis.

**Funding acquisition:** Maureen McEvoy, Julie Luker, Lucy K Lewis.

**Investigation:** Maureen McEvoy.

**Methodology:** Maureen McEvoy, Julie Luker, Caroline Fryer, Lucy K Lewis.

**Project administration:** Maureen McEvoy.

**Resources:** Maureen McEvoy, Caroline Fryer, Lucy K Lewis.

**Supervision:** Maureen McEvoy.

**Validation:** Maureen McEvoy, Julie Luker, Caroline Fryer.

**Visualization:** Maureen McEvoy, Julie Luker, Caroline Fryer.

**Writing – original draft:** Maureen McEvoy, Julie Luker, Caroline Fryer, Lucy K Lewis.

**Writing – review & editing:** Maureen McEvoy, Julie Luker, Caroline Fryer, Lucy K Lewis.

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
