## [Decision Letter · Decision Letter 0]

23 Jul 2020

PONE-D-20-07294

Changes in physiotherapists perceptions of evidence-based practice after a year in the workforce: a mixed-methods study

PLOS ONE

Dear Dr. McEvoy,

Thank you for submitting your manuscript to PLOS ONE. After careful consideration, we feel that it has merit but does not fully meet PLOS ONE’s publication criteria as it currently stands. Therefore, we invite you to submit a revised version of the manuscript that addresses the points raised during the review process.

Both reviewers highlighted major points that needed to be clarified with respect to the methodology, and I concur that these points are important in enhancing the our understanding of the study, and our confidence with the findings being reported.

We look forward to receiving your revised manuscript.

Kind regards,

Catherine M. Capio

Academic Editor

PLOS ONE

Journal Requirements:

Reviewers' comments:

Reviewer's Responses to Questions

**Comments to the Author**

1. Is the manuscript technically sound, and do the data support the conclusions?

Reviewer #1: Partly

Reviewer #2: Yes

2. Has the statistical analysis been performed appropriately and rigorously? 

Reviewer #1: Yes

Reviewer #2: Yes

3. Have the authors made all data underlying the findings in their manuscript fully available?

Reviewer #1: Yes

Reviewer #2: Yes

4. Is the manuscript presented in an intelligible fashion and written in standard English?

Reviewer #1: No

Reviewer #2: Yes

5. Review Comments to the Author

Reviewer #1: OVERALL REVIEW

The contribution to knowledge gap of this research report is undeniable. There are hardly any studies on evidence uptake after having learned the principles of EBP in entry-level education particularly in Physiotherapy. Therefore it is important that the findings of this research be communicated with clarity so that readers and follow through the data collection procedures, data analysis and insights of the authors.

Major concerns are the following:

Background/introduction. The objective depicts a qualitative study when the entire paper talks about mixed- methods. Provide an objective that would justify quantitative data collection and process.

Design. Explain more clearly how qualitative and quantitative procedures were used for the study. Was one dependent on the other? If qualitative data was used to support quantitative data, how and when was the qualitative arm of the study done? Did the authors analyzed the quantitative findings in order to direct or focus their interview questions? Or did they ask the interview questions independent of the quantitative results?

Description of participants; who was the target population, how many were they? What are the inclusion and exclusion criteria?

How was quantitative data treated? The authors mentioned that only respondents who answered both sets of questionnaire in Times 1 and 2 were included. Yet there are respondents who did not answer all items as expressed in lines 182-184? ‘there was not complete matched data from all participants for each of the six EBP domains’?

The authors stated that a limitation of this study is the method of analysis for the qualitative data. It puts into question the results of the study; why wasn’t an “in-depth” analysis conducted when that was the intention for doing a qualitative arm as stated in the design? This has to be clarified in order to align methodology and results.

Discussion. Most of the concepts were not followed through and requires further explanation and insights from the authors. This is very important if the results of the studies are to be useful for readers.

Minor to moderate concerns are the following:

Sentence and paragraph construction could be greatly improved for readability and clarity.

Formatting of tables for ease of reading the results and findings.

I gave some suggestions that could be useful but authors are free to work around this.

Addressing these concerns could make this manuscript more acceptable. Please find below specifics of my general review:

ABSTRACT

Line 22. May I suggest to use physiotherapy participants or simply participants, instead of physiotherapy students since they were not students at the time of the study.

BACKGROUND

Lines 79 to 80. I suggest to include an aim or objective that would pertain to the quantitative aspect of the research in order to anticipate the collection of quantitive data. Eg. To determine changes in EBP profile…

METHODS

Design

Line 94, remove period after workforce since the sentence is not yet completed.

How did the researchers treat the two methods for data collection? Was one dependent on the other or were they treated independent of each other. Were the interviews conducted taking into consideration the changes in scores of the quantitative data?

Participants

Line 106, Is it possible that one can complete entry-level of training in university and not graduate? If completion of entry-level training subsequently leads to being ‘graduated' then you can delete the words ‘and graduated’ to go straight to ‘and entered the physiotherapy workforce…’.

Are there exclusion criteria for the participants? (I think you mentioned these in the results part. You can mention it instead here).

Lines 107 - 108. Consider revising the sentence, starting with your intention to achieve a 80% power. Example. In order to achieve a power of 80% for the quantitative data……a minimum of 34 participants were needed.

Or: A minimum number of 34 participants was needed for the quantitative data set in order to obtain an 80% power for a two-tailed distribution, with alpha set at 0.05 and a medium effect size (d=0.5).

Procedure

Line 112. Consider improving the sentence construction in describing the questionnaires for easier reading and comprehension. Example:

Quantitative data were collected using two questions, the EBP2 and the KREC. The EBP2 has been psychometrically tested and shown to be valid and reliable etc. It may not be necessary to state that it is developed, since it has been used several times already. Then move on to describe its features; number of items, domains etc.

Line 119. Is 'actual knowledge’ used as a proper noun here? also why is it italicized? When using the word "subsequent", do you mean it was answered AFTER the EBP2? Was this an important or significant step in the process? Would it have mattered if participants answered K-REC first before EBP2? If the sequence in which the questionnaires were answered does not matter, then no need to write the word ‘subsequent'. If the sequence mattered, kindly give a brief explanation for this. The K-REC questionnaire has nine items that measures a participants knowledge of EBP using… It has been found to be valid and reliable….etc.

Line 123. Use an active voice when possible. Example: Participants completed the questionnaires at two timepoints; 1) upon their completion of entry-level training, Time 1, and 2) one year after entering the workforce, Time 2. At time point 1, participants answered the questionnaires using a pen-and-paper format. For time point 2, participants answered using an electronic version of the questionnaire through Survey Monkey, due to the participants' being geographically dispersed

Suggestion. How about using Time 1 and Time 2 instead of Timepoint 1, Timepoint 2.? If you agree, please revise throughout the manuscript.

Lines 131 to 133. When you say that “feedback from a reference group of six ….”, were you talking about content validation of the interview guide or did you pilot test the interview guide with this reference group? Also, my downloaded copy of the manuscript did not include S1 Appendix being referred to at this part.

Line 136. Under Design, I commented/recommended that you state the relationship of the two methods of data collection. It might be better to place lines 136 - 138 in that area of the paper. So that this part of the paper, similar to quantitative data collection, will discuss data collection tools and data collection process.

So instead, you can proceed by describing the interview guide in terms of the topics covered.

Line 141 onwards. An interviewer, independent of the research team and experienced in qualitative interviewing, conducted telephone interviews with the participants.

DATA MANAGEMENT AND ANALYSIS

Line 148 onwards. Suggestion: The predictive Analytic Software (PASW) Statistics 17.0 was used for quantitative data analysis. Only completed matched data (timepoints 1 and 2) were included for analysis.

Consider breaking down sentences when conveying different ideas or thoughts. Eg. Separate the sentences when talking about Likert scores and the fact that the domains have different number of items. Example: The Likert-scores for the EBP2 were treated as interval data. Then move on to describe the domain score.

Is there something more that you would want to discuss in reference to the different domain scores? Did this matter in data analysis. Because if this sentence (lines 150-152) just means to say that the scores will be different, it can be derived from the earlier description of the instrument, wherein the number of items for each domain was listed. But if this was transformed to make them ‘equal' for data analysis, then maybe this should be describe.

Line152. I don’t think it is necessary to include ‘for Actual Knowledge' to this. The K-REC gives a maximum score of 12. The participants responses were sacred using the set scoring guidelines.

Line 154. delete 'Actual Knowledge domain’

Line 160 to 168. I think this paragraph can be refined by using an active voice; state clearly what steps were taken for analysis; then towards the end describe the entire process as iterative so that it does not have to be stated repeatedly in the paragraph.

Two researchers (CF and JL) independently (or separately) analyzed the qualitative data using a staged process of thematic analysis. First researchers read the transcripts repeatedly to familiarize themselves with the data. Second, codes were inductively allocated to small sections. Third, comparison across transcripts and codes were done, inductively grouping the codes in a meaningful manner to form categories and themes within each category. This was an iterative process requiring the researchers to meet and discuss regularly to eventually reach a consensus.

In reference to line 163: Do the researchers think that analysis using software and hard copies affected the data? If so, maybe a short explanation is warranted. If not, this then can be deleted.

RESULTS

QUANTITATIVE

May I suggest to use 'completed an entry-level physiotherapy program’ as descriptors for participants to veer away from their ‘student’ status. This would help in focusing the readers, that the study is following through working physios instead of students.

The count of the participants is hard to follow through but it would be helpful if you keep in mind your target participants (physios who competed their final year of entry-level training at USA in 2013). So start with this number (how many were they actually). From this number, subtract those that copy with your exclusion criteria (plan to work overseas, not work in physio and did not consent to participate after one year). I think you will end up with 65 eligible participants. Then continue to discuss what happened to these eligible participants and how many viable data were left. Coming up with an illustration or figure for this might be helpful for the reader to follow.

At this point, I am confused as to how many viable data were analyzed? What do the researchers mean by lines 182-184? ‘there was not complete matched data from all participants for each of the six EBP domains’? how then was this treated?

Best also to describe participants at timepoints 1 and time point 2.

At Time 1, 50 participants were mostly female with a mean age of 22 +/- 4 (range 20 - 47 years)

At Time 2, 14 participants were interviewed, there were 10 females, with an average of 22 +/- 1 year (range 20 to 23).

I think it is important to give characterization as to their workplace or work setting. This could give a better context of the succeeding discussion of results.

Table 1 can be further improved for easy reading. Separate the cells for Change “findings”; one cell each for raw score difference, p-value and effect size. You can also put as footer for the table, the meaning of Time 1 (completion of entry-level training) and Time 2 (one year working?). The title of table can also be improved. It should be able to answer the following; what is being presented? how is the data classified? Indicate number of data/participants. It might help if you look at other tables

Figure 1 is a good illustration to show the change. It could be more informative if the data re: ES, p-values be shown there as well. Eg. Relevance (p-<.001, ES = .69)

Lines 207-209. Elaborate and describe a bit more what happened to the maximum domain scores. i believe this could be better situated and described under data management and analysis.

QUALITATIVE

I think the results should answer the aim or objective of the research as indicated in the background; "To explore the changes in EBP knowledge, attitudes and behaviors from graduation to one year in workforce of a cohort of physiotherapists."

Therefore, just present the themes gathered that best represents this aim. It is unnecessary to discuss how the other categories will be treated if it has nothing to do with the current paper. I think this adds to confusion and questions regarding the paper.

I would suggest further improving your table that shows qualitative findings. Tables are meant for the reader to see at a glance the most significant findings of a study. Having 12 pages of this makes it difficult to do that.

My suggestion…include samples of quotations from the respondents in the text instead of put them on the table. For the table, you may reformat to just include the themes and sub-themes.

Example:

Example in text:

Most participants reported using EBP skills and knowledge in their first year to inform their clinical reasoning, empower clients with knowledge or learn from the clinical expertise of experienced colleagues. Some example to attest to these are:

"So I’ve seen a few patients that I was having trouble with, searched some evidence, had a look at some new treatment techniques and stuff like that and obviously searched if they were significantly… sort of if they were going to work and tried them clinically and they did

work, which was very helpful." (Participant 14 for informing clinical reasoning)

"Look I’ve had a few patients that they may have been a little bit apprehensive about some of the treatments that we use but then we sort of explain to them about the evidences behind it and it really does help them to feel confident in what we’re doing and that we’re not

just doing some sort of weird treatment, that we actually have got evidence behind it and we’ve done research into it." (Participant 56, Empowering patients)

When using this format, the reader is easily referred to sample quotations without having to go back to the table.

This is a suggestion. There may be other ways to improve this. The main goal is to make the table an easy reference point.

DISCUSSION. (No line numbers in this part)

Paragraph 2.

Most of the ideas presented in this paragraph pertaining to relevance needs to be elaborated on. For example, the concept that since there is a large quantity of MS research then it is perceived to be more relevant? What is the basis for saying this? Similarly the sentence/concept related to wider health care system and funding model of direct care need explanation and elaboration to better understand how this is related to decrease in Relevance domain of physios in the workplace. Finally the last concept there on competing demands of improving clinical skills…etc. needs further explanation on how it affected perception of relevance for EBP.

Paragraph 3. On reduced confidence.

Improving sentence construction of this could help in delivering the message clearly.

Eg. The overall decline in the Confidence domain of the participants may have been influenced by their reported barriers to use EBP. These barriers would lead to less frequent practice of looking up evidence and thus possibly decrease confidence. Similarly, McEvoy’s study showed that….

You mentioned that McEvoy [13] study did not use qualitative data to explore…how then do you think your data is better or different from this study and how does this current study better explain the decrease in confidence?

Paragraph 4. on workplace culture.

Shifting to an active voice could better deliver the message. Ex.

Results of this study indicate that a workplace culture that values research evidence strongly supports EBP use and helps in maintaining/improving ? confidence. Some of the practices that cultivate this culture include….(state here).

Paragraph 5. Transition to workplace

Is this all about PEAK? Maybe just give a summary of what this is about and then provide your own insight about this? Maybe there are other options?

Paragraph 6. Recommendations.

It might be helpful to organize your thoughts for this…recommendations in education and recommendation in practice? (you referred to this in your introduction, line 77). Did the data lead you to believe that improvement in EBP education in entry-level could help maintain the Profile domain scores? You mentioned that there are several recommendations to be considered but actually only mentioned the ‘re-fresh’ training. It might help if you outline the recommendations and discuss each.

Paragraph 7. Limitations.

No need to state strengths, just disclose limitations. I question your second limitation; "of using descriptive thematic analysis and not allowing for rich in-depth exploration of the findings”. Your purpose for collecting qualitative data is to gain a deeper and broader understanding (line 94-95). You further described the use of an interpretive paradigm to allows rich description of lived experiences. Your description of qualitative data analysis emphasized an iterative approach to make sure that nothing is left undiscussed and undecided. Yet you say that this is not a rich in-depth exploration of the findings? This limitation does not follow through with your intention in using qualitative analysis.

Your third limitation also puts to question the validity of the quantitative data. What is the objective of your study? Did you aim to determine how these factors influence the data? (I mentioned that you should also include an objective for your quantitative arm in the introduction).

CONCLUSION.

This part should go back to the research questions/ or research objective as stated.

One year into working, physiotherapists showed decline in all EBP profile domains, most signicantly for…. In the first year of work, the participants experience of EBP revolved around the themes of…

Reviewer #2: This is a well written manuscript with the authors evidencing their expertise and research capacity spanning many years in evidence-based practice and development of EBP competences for entry level health professionals. The authors ask an interesting question: do perceptions of evidence based practice change during the transition from physiotherapy student to entry level clinician? This is an important topic as EBP is now a core component and often statutory requirement in entry level professional programmes but whether and how effectively this translates to autonomous professional practice is not clear.The authors adopt a mixed methods approach to allow broad insight in this area.

It would however, be beneficial to identify for the reader whether a convergent or sequential (explanatory/exploratory) design was taken for this mixed methods approach. This would help to situate the qualitative work better in the study design. While the qualitative methodology describes a staged inductive approach to thematic analysis, the descriptive approach used to report results leads to many (maybe too many) sub-themes which tended to match back to the questions asked. To this end it would be helpful to know more about the design of the question schedule and whether the quantitative data findings informed this process. Overall descriptive thematic analysis would not be my preferred choice as the results generating 9 themes with multiple sub-themes made for lengthy and difficult reading with limited meaningful synthesis to take away. In the qualitative results table some participants are labelled xx and this needs to be addressed to assure fidelity of data.

With respect to the quantitative study, more detail relating to the data (sd etc) and sources of this data used in the power calculation would be helpful for readers. The authors highlight that the EBP questionnaire which uses Likert scales would be treated as interval data, however the same information and justification for parametric testing of the K-REC instrument is not provided. Similarly in the reported results, it would be better to address the K-REC results separately as it is a different construct with different psychometric properties to those in the EBP profile questionnaire.

The discussion section adopted a very narrow approach to the findings, lacking higher level discourse which was disappointing. In fact only two new references to the literature were introduced at this stage. Acknowledging the paucity of studies directly related to this research question, I would have expected to see a wider discussion in relation to the qualitative literature addressing transitioning to professional practice following graduation and similarities with the findings in this study. Similarly, I would have expected reference to stages of learning as novice, advance beginner, proficient, competent, and finally expert where ones does see greater confidence in novices than beginner/proficient learners as recognition of contextual factors and greater insight into deficits in knowledge become more apparent.

One minor consideration that was confusing was that in the intro pg4 line 65 they cite McEvoy et al [13] as providing workplace transition qualitative data and in the discussion paragraph 3 they cite the same study as providing longitudinal data that lacks qualitative insights.

6. PLOS authors have the option to publish the peer review history of their article (what does this mean?). If published, this will include your full peer review and any attached files.

Reviewer #1: No

Reviewer #2: **Yes: **Olive Lennon

---

## [Author Response · Author response to Decision Letter 0]

31 Oct 2020

1. The authors have attempted to ensure the manuscript meets PLOS ONE's style requirements, including those for file naming.

2. Data availability has been complied with. Data not included within the manuscript, has been provided in Supporting Information

---

## [Decision Letter · Decision Letter 1]

7 Dec 2020

Changes in physiotherapists perceptions of evidence-based practice after a year in the workforce: a mixed-methods study

PONE-D-20-07294R1

Dear Dr. McEvoy,

We’re pleased to inform you that your manuscript has been judged scientifically suitable for publication and will be formally accepted for publication once it meets all outstanding technical requirements.

Kind regards,

Catherine M. Capio

Academic Editor

PLOS ONE

Additional Editor Comments (optional):

Reviewers' comments:

Reviewer's Responses to Questions

**Comments to the Author**

1. If the authors have adequately addressed your comments raised in a previous round of review and you feel that this manuscript is now acceptable for publication, you may indicate that here to bypass the “Comments to the Author” section, enter your conflict of interest statement in the “Confidential to Editor” section, and submit your "Accept" recommendation.

Reviewer #1: All comments have been addressed

Reviewer #2: All comments have been addressed

2. Is the manuscript technically sound, and do the data support the conclusions?

Reviewer #1: Yes

Reviewer #2: Yes

3. Has the statistical analysis been performed appropriately and rigorously? 

Reviewer #1: Yes

Reviewer #2: Yes

4. Have the authors made all data underlying the findings in their manuscript fully available?

Reviewer #1: Yes

Reviewer #2: Yes

5. Is the manuscript presented in an intelligible fashion and written in standard English?

Reviewer #1: Yes

Reviewer #2: Yes

6. Review Comments to the Author

Reviewer #1: This version of the manuscript now shows coherence of the objectives, methods and discussion. It is well-written such that it is easy to understand and follow. After reading the manuscript, I reviewed the comments I had previously given and the authors were able to adequately respond and act on them. Congratulations to the authors for this work!

Reviewer #2: I am satisfied that the authors have adequately addressed the reviewers' comments and the manuscript is improved.

7. PLOS authors have the option to publish the peer review history of their article (what does this mean?). If published, this will include your full peer review and any attached files.

Reviewer #1: No

Reviewer #2: **Yes: **Olive Lennon

---

## [Editor Report · Acceptance letter]

11 Dec 2020

PONE-D-20-07294R1 

Changes in physiotherapists’ perceptions of evidence-based practice after a year in the workforce: a mixed-methods study 

Dear Dr. McEvoy:

I'm pleased to inform you that your manuscript has been deemed suitable for publication in PLOS ONE. Congratulations! Your manuscript is now with our production department. 

Kind regards, 

on behalf of

Dr. Catherine M. Capio 

Academic Editor

PLOS ONE